# Protective Effects of Fucoidan on Iodoacetamide-Induced Functional Dyspepsia via Modulation of 5-HT Metabolism and Microbiota

**DOI:** 10.3390/ijms26073273

**Published:** 2025-04-01

**Authors:** Tianxu Liu, Muyuan Ma, Yonglin Wu, Ismail Muhammad Asif, Daosen Chen, Lichong Liu, Minghui Zhang, Yijie Chen, Bin Li, Ling Wang

**Affiliations:** 1College of Food Science and Technology, Huazhong Agricultural University, Wuhan 430070, China; liutianxu1999@126.com (T.L.); mmy19980818mmy@163.com (M.M.);; 2Key Laboratory of Environment Correlative Dietology (Huazhong Agricultural University), Ministry of Education, Wuhan 430070, China; 3Tianjin Key Laboratory of Food Science and Health, School of Medicine, Nankai University, Tianjin 300071, China

**Keywords:** fucoidan, functional dyspepsia, visceral hypersensitivity, 5-HT, microbiota

## Abstract

As the major polysaccharide in brown algae, fucoidan possesses broad biological abilities and has been reported to improve gastrointestinal health. Functional dyspepsia, a common non-organic disease, is a complex of symptoms mainly characterized by pathogenesis, such as visceral hypersensitivity, gastric dysmotility, and inflammation. To date, the effects of fucoidan in regulating functional dyspepsia with visceral sensitivity remains unclear. In the current study, iodoacetamide was employed to establish a mouse model of visceral hypersensitivity. Meanwhile, fucoidan was orally administrated for fourteen days. Indicators were conducted to evaluate the potential of fucoidan as the ingredient of complementary and alternative medicine for functional dyspepsia, such as levels of serum hormones, expression of receptors, and gut microbial profile. The results show that oral administration of fucoidan led to significant reductions in the secretion of 5-hydroxytryptamine, cortisol, and corticosterone. Additionally, it decreased the expression of 5-hydroxytryptamine_-3_ receptors, with regulation of 5-hydroxytryptamine metabolism and improvement of gut microbial imbalance. The above results suggest fucoidan could ameliorate visceral hypersensitivity by modulating 5-HT metabolism and microbiota. The current findings indicate that fucoidan has potential as a biological component in the adjuvant treatment of functional dyspepsia and for its expanded utilization in the food and medical fields.

## 1. Introduction

Functional dyspepsia (FD) and irritable bowel syndrome (IBS) are categorized under functional gastrointestinal disorders (FGIDs) and are the foremost prevalent subgroups within this classification. Among them, FD is a chronic gastrointestinal disease characterized by a non-organic pathology during upper gastrointestinal endoscopy [1]. Unlike other organic gastrointestinal disorders, such as inflammatory bowel disease (IBD), individuals afflicted with FD typically exhibited no obvious organic pathology. Recently, Lee K. et al. conducted a meta-analysis, which reported that the global pooled prevalence of FD is approximately 6.8% according to Rome IV criteria [2]. The primary symptoms of FD include fullness, early satiety, epigastric pain, abdominal burning, loss of appetite, nausea, which have numerous adverse impacts on the quality of patients [3]. The Rome IV consensus divides FD into two subgroups based on the symptoms, postprandial distress syndrome (PDS), characterized by fullness and flatulence, and epigastric pain syndrome (EPS), presenting with abdominal pain and burning [4]. The pathogenesis of FD is relatively complex with many influencing factors, involving gastrointestinal dysmotility, visceral hypersensitivity, disordered brain–gut interaction, injured intestinal microenvironment, etc.

In view of the intricate pathogenesis and symptomatology of FGIDs, tradition drug intervention remains challenging in managing various discomfort of FD. A growing number of FD patients are turning to complementary and alternative therapies to relieve their symptoms. Ganguli S. et al. conducted a community study and found that nearly 52.5% of gastroenterology outpatients employed alternative medicine [5]. Especially, among patients with FGIDs, the prevalence of alternative medicine usage was even higher, with dietary/herbal interventions being the most commonly utilized approaches [6]. Marine plants encompass a vast array of members, including 13 phyla and over 10,000 species, which not only serve pivotal roles in ecosystem equilibrium but also provide shelter and sustenance for numerous organisms [7]. These years, the burgeoning global health product industry has increasingly focused on marine bioactive compounds, which exhibit broad biological activities, such as antibacterial, anti-inflammatory, and analgesic properties [8]. Algae constitute 90% of marine plants, with brown algae representing a predominant group, comprising 66% of total algae consumption [9,10]. In the food industry, brown algae were initially utilized as a stabilizer and thickener; however, its rich and distinctive biological activity has garnered increasing attention recently [11,12]. Notably, brown algae possess a highly abundant polysaccharide content, comprising over 50% of its total dry weight [13].

Fucoidan (FUC), one of the most important bioactive polysaccharides in brown algae, exhibits remarkable biofunctions due to its unique structure and rich content of the sulfuric acid group. These functions encompass anti-tumor, anti-inflammatory, antioxidant, immune-regulatory properties, etc. [14,15], thereby indicating that FUC may have potential in modulating dysfunction of multi-systems. To date, FUC has shown beneficial regulatory effects on the gastrointestinal tract according to previous studies. By modulating immune response, reducing inflammation, and mitigating oxidative stress, FUC could effectively ameliorate gut microenvironment disturbances under various pathological conditions, such as IBD and colorectal cancer [16,17]. In addition, it seems that FUC may have the potential to improve muti-system disorders, including the nervous system [18]. Nonetheless, whether FUC has positive impacts on visceral hypersensitivity remains unknown.

According to our previous studies, which indicated that laminarin can partly alleviate FD with visceral hypersensitivity, and as another major polysaccharide in brown algae, we hypothesized that the intervention of FUC could temper visceral hypersensitivity [19]. In this study, the effects of FUC on iodoacetamide (IA)-induced visceral hypersensitivity were investigated using molecular biological tools, and a preliminary exploration of the underlying mechanisms was conducted [20]. These findings offer an ingredient for use in supplementary treatment medicine for FD patients with visceral hypersensitivity as the main pathogenesis and provide a theoretical basis and new insights for the utilization of FUC in food and medical fields.

## 2. Results

### 2.1. Effect on Serum 5-HT, ACHT, COL, and CORT

The concentrations of serum 5-hydroxytryptamine (5-HT), adrenocorticotrophic hormone (ACHT), cortisol (COL), and corticosterone (CORT) were determined using ELISA kits. Compared to the control check (CK) group, mice in the model control (MC) group exhibited significantly higher levels of serum 5-HT, COL, and CORT (*p*
< 0.05, Figure 1B,D,E). However, the concentration of ACHT did not differ between these two groups (Figure 1C). The imbalance of these biochemical molecules indicates a condition of stress and depression induced by IA (Appendix A). Low-dose FUC (FUCL) intervention greatly reduced the secretion of 5-HT, ACHT, and CORT, while oral high-dose FUC (FUCH) decreased levels of 5-HT, COL, and CORT (*p*
< 0.05). Pinaverium bromide failed to regulate the above hormones and transmitters.

### 2.2. Effect on the Expression of the 5HT_3_ Receptor in Stomach, Duodenum, and Brain

The 5-HT_3_ receptor (5-HT_3_R) has been revealed to be responsible for the progression and persistence of visceral hypersensitivity via the serotonergic pathways [21]. In this section, the expression of 5-HT_3_R at both the protein and mRNA levels was evaluated in the stomach, duodenum, and brain of the mice in each group. As shown in Figure 2A–C, at the protein level, IA administration led to the overexpression of 5-HT_3_R in the duodenum of mice in the MC group (*p*
< 0.05), while no significant changes were observed in the stomach and brain. FUCH administration greatly reversed the alterations in the duodenum (Figure 2B); in addition, oral FUCL reduced 5-HT_3_R protein expression in the stomach (Figure 2A). However, pinaverium bromide did not exhibit a similar ability. At the mRNA level, oral IA induced obvious overexpression of the gene encoding 5-HT_3_R in the stomach (*p*
< 0.05, Figure 2D). The levels of gene encoding 5-HT_3_R did not differ between the CK and MC groups in the duodenum, hippocampus, and hypothalamus (Figure 2E,F_1_,F_2_). Meanwhile, FUCL, FUCH, and pinaverium bromide intervention not only reduced the expression of the gene encoding 5-HT_3_R in the stomach but also in the hippocampus and hypothalamus (*p*
< 0.05, Figure 2D,F_1_,F_2_).

### 2.3. Effect on the Expression of Key Genes in the Duodenum and Brain

The interaction between the gut and brain is tightly associated with visceral hypersensitivity including 5-HT metabolism and sensation transduction [22]. To verify this opinion, the expression levels of the genes encoding serotonin transporter (SERT), tryptophan hydroxylase 1 (TPH1), tryptophan hydroxylase 2 (TPH2), transit receptor potential cation 4 (TRPC4), and paired box 4 (PAX4) in the duodenum and brain were determined by RT-qPCR. The alterations of the expression of these genes were primarily concentrated in the hippocampus. The IA intervention obviously increased the expression of genes encoding TPH2 and TRPC4 in the hippocampus (*p* < 0.05, Figure 3F,G). Oral FUCL, FUCH, and pinaverium bromide reversed the above changes greatly (*p* < 0.05). In addition, FUC elevated the expression of PAX4 as well (*p* < 0.05, Figure 3H). However, no significant changes in gene expression could be observed in the duodenum of mice in each group (Figure 3A–D,I–L).

### 2.4. Effect on Gut Microbiota

#### 2.4.1. Microbial Diversity and Richness

In this study, 16S-rRNA sequencing analysis was employed to determine the gut microbial profile. As shown in Figure 4, the α-diversity and β-diversity were compared between mice in each group. The Ace, Chao, Shannon, Sobs indexes were used for the description of the α-diversity of the gut microorganisms. The α-diversity of the gut microbiota showed no significant difference between the CK and MC groups. The intervention of FUC did not impact the microbial α-diversity, as indicated by the results of the Ace, Chao, Shannon, and Sobs indexes (Figure 4A–D). Meanwhile, principal coordinates (PCoA) and partial least squares discriminant (PLS-DA) analysis were employed for the evaluation of the β-diversity at the OTU level. According to the results of the PCoA analysis, the structure of the gut microbiota did not differ between each group (Figure 4E). Whereas the PLS-DA analysis suggested that the profile of the mice in the MC group was significantly different compared to the CK group, possibly due to the administration of IA (Figure 4F). The interventions of FUCL, FUCH, and pinaverium bromide ameliorated the above changes to varying degrees, with FUCH showing particularly notable improvement. These results indicate that FUC may possess beneficial effects in regulating the gut microbial diversity and richness of FD mice induced by IA.

#### 2.4.2. Microbial Composition

Subsequently, the composition of the gut microorganisms was determined at the phylum and genus levels (Figure 5). At the phylum level, the predominant bacteria of the microbiota were Bacteroidota, Firmicutes, Verrucomicrobiota, and Patescibacteria across all groups (Figure 5A). Their abundance changed obviously after the intervention. The administration of IA greatly reduced the richness of Firmicutes and Patescibacteria (*p* < 0.05, Figure 5C,D). FUC significantly recovered the reduced colonization of Patescibacteria, while FUCH specifically elevated the percentage of Firmicutes in the gut bacteria as well (*p* < 0.05). At the genus level, *Muribaculaceae*, *Akkermansia*, *Odoribacter*, *Alistipe*, *Lachnospiraceae_NK4A136_group*, and *Candidatus_Saccharimonas* were the most colonized bacteria across all groups (Figure 5B). Compared to mice in the CK group, IA intervention greatly increased the abundance of *Muribaculaceae* (*p* < 0.05, Figure 5E,F). Both FUCL and FUCH administration decreased the richness of *Muribaculaceae* (*p* < 0.05). Oral pinaverium bromide did not exhibit similar effects compared to FUC at the phylum and genus levels. These findings demonstrate the regulatory abilities of FUC, particularly FUCH, in modulating gut microbial composition disorders.

### 2.5. Correlation Between the Gut Microbial Composition and Visceral Hypersensitivity-Related Indicators

The correlations between the gut microbiota and visceral hypersensitivity-related indicators were established using spearman’s analysis at both the phylum and genus levels (Figure 6). At the phylum level, representative results include the following (Figure 6A): The abundance of Firmicutes was negatively associated with the levels of serum COL and CORT (*p* < 0.05). The richness of Proteobacteria was positively related to the expression of the gene encoding 5-HT_3_R in the hippocampus and hypothalamus (*p* < 0.05 or 0.01). At the genus level, the abundance of *Muribaculaceae* was positively associated with the expression of the gene encoding TPH2 in the hippocampus (*p* < 0.01). The level of serum COL was negatively related to the richness of *Alistipe*, *Lachnospiraceae_NK4A136_group*, *Enterococcus*, and *Clostridia* (*p* < 0.05 or 0.01 or 0.001).

## 3. Discussion

FD is a prevalent kind of FGID characterized by complex pathogenesis and symptomatology. Visceral hypersensitivity has been observed in a large proportion of patients with FD, and it is suggested to be related to the altered communication of the gut–brain axis [23], annoying a number of patients. Traditional clinical medicine, including proton pump inhibitors, histamine-2 receptor antagonists, and dopamine receptor antagonists, may not adequately relieve gastrointestinal discomfort in FD patients [24]. Therefore, an increasing number of patients seek complementary and alternative treatment for holistic relief of their symptoms. Among them, the herbal/dietary supplement exhibited the most potential efficacy. According to our pervious study and current evidence, FUC has remarkable beneficial abilities in relieving dysfunction involving multi-systems [25]. Therefore, we hypothesized that FUC could be an effective and safe ingredient in complementary and alternative medicine for FD treatment. In this study, the levels of serum hormones, 5-HT metabolism-related markers, and changes in the gut microbial profile have all been evaluated to verify the above opinions.

A mouse model of visceral hypersensitivity was established according to Ji E. et al.’s description [26]. IA, a protease inhibitor, may induce an acute lesion in visceral sensation with depression- and anxiety-like behaviors [27]. In our study, the level of some indicators did not differ from mice in the CK and MC groups, which may be attributed to the milder and shorter duration of our modeling methods. These features align with the characteristics of FD, namely, upper gastrointestinal dysfunction without organic changes [28]. In addition, previous studies revealed that IA administration disturbs the balance of both the gut microenvironment, central nervous system, autonomic nervous system, etc. [29]. This animal model can be used for a comprehensive evaluation of the potential of FUC as an ingredient in complementary and alternative medicine.

The levels of serum 5-HT, ACHT, COL, and CORT were determined in this study. 5-HT (serotonin), a vital gastrointestinal neurotransmitter, is responsible for transmitting information and sensation between nerve cells [30]. Researchers have elucidated the relationship between 5-HT metabolism and FGIDs, which encompasses the regulation of gastric motility, visceral sensitivity, appetite, pain perception, etc. [31]. Abnormal activation of 5-HT may contribute to visceral hypersensitivity and chronic inflammation in multi-systems [32]. According to our results, the introduction of IA elevated the concentration of serum 5-HT, which is suggested to be related to depression-like symptoms under visceral hyperesthesia. Increasing evidence has revealed the neuroendocrine pattern under acute and chronic stress of FGIDs. With the key role of brain–gut axis disorders in visceral hypersensitivity revealed, the above concepts are receiving more attention. ACHT, COL, and CORT are secreted by pituitary and adrenal glands, the responsiveness of the hypothalamic–pituitary–adrenal axis usually increases during states of stress [33,34]. We observed that IA upregulated the level of serum CORT. Winston J. et al. reported that CORT may contribute to neuroinflammatory injury, which participates in the development of visceral hypersensitivity [35]. Although not fully tested, the metabolism of glucocorticoids could potentially be related to the dysregulation of the brain–gut axis. Surprisingly, FUC showed abilities in downregulating ACHT, FUCL, and FUCH compared to pinaverium bromide. Hou L. et al. demonstrated that the auricular vagus nerve’s stimulation is effective in relieving FD symptoms by inhibiting hypothalamus–pituitary–adrenal axis [36]. ZhiShiXiaoPi tang appears to exhibit similar effects via decreasing oxidative stress [37]. Additionally, electroacupuncture modulated the secretion of ACHT and CORT and suppressed low-grade inflammation in FD rats [38]. Therefore, we speculate that FUC may influence overactivation of the pituitary–adrenal axis during visceral hypersensitivity by reducing inflammation and oxidative stress.

The association of 5-HT and its receptors has been extensively discussed, with 5-HT_3_R as one of the most pivotal receptor subtypes [39]. 5-HT_3_R is broadly distributed throughout the body, including the gastrointestinal tract, hippocampus, and various nervous or non-nervous tissues [40]. When stimulated by 5-HT, 5-HT_3_R on the central terminal of the vagal afferent nerve enhances the transmission of glutaminergic synapses to secondary neurons in the solitary nucleus in the brainstem [41]. Lyubashina O. et al. carried out relevant research and addressed the 5-HT_3_R-dependent mechanisms in the persistence of visceral hypersensitivity and hyperalgesia [21]. In this study, the expression of 5-HT_3_R in multi-tissues was determined at both the protein and mRNA levels. The results suggested IA administration significantly increased sensory disorders of the gastrointestinal tract. Notably, FUC showed a remarkable 5-HT_3_R antagonist-like ability in the hippocampus and hypothalamus at the mRNA level. Previous studies have proven that the dysfunction and abnormal 5-HT metabolism in the hippocampus and hypothalamus are associated with depression and visceral hypersensitivity [32,42]. Song M. et al. emphasized the regulatory effects of FUC on 5-HT_3_R expression in the gastrointestinal tract [43]. Our findings not only corroborate their opinion but also unveil the beneficial influence of FUC on the overexpressed 5-HT_3_R in the brain.

SERT and TPH1/TPH2 are the transporter and key precursor synthetases of 5-HT. SERT is a protein widely expressed throughout the body, characterized by its high affinity for 5-HT. It regulates the level of 5-HT in the systemic circulation through the reuptake of 5-HT [44]. TPH1, predominantly expressed in enterochromaffin cells (non-neuronal cells), is responsible for the conversion of tryptophan to 5-HT. Conversely, in neurons of the brain and plexus, TPH2 acts as the rate-limiting enzyme [45]. Surveys conducted by Yuan J. and Jun S. et al. revealed the association between the polymorphism of the genes encoding SERT, TPH1, and TPH2 and the severity of IBS [46,47]. This emphasizes the connections between 5-HT metabolism and sensory dysfunction in FGIDs. The findings of the current study suggest FUC could reverse the reduced expression of the gene encoding SERT and increased the gene encoding TPH2 induced by IA in the hippocampus. Recently, Li X. et al. reported that costunolide, an herbal extract, exhibited similar regulatory effects [48]. By depressing the activity of mastocyte and the metabolism of 5-HT in the central nervous system (CNS), costunolide greatly relieved depressive behavior and intestinal dysfunction in stress-induced IBS mice. The anti-inflammatory ability of FUC may be the reason why it showed above beneficial effects.

TRPC4 represents a subclass of Ca^2+^-permeable non-selective cation channel protein, which is distributed throughout the body, with notable abundance in the limbic region of the cerebral cortex [49]. Though the relationship between TRPC4 and visceral hypersensitivity remains unclear, recent studies have implicated its hyperexcitability in the neurons of patients with epilepsy. Furthermore, it has been observed that several antidepressant drugs exert their effects by suppressing the expression of this receptor [50]. PAX4 is integral for the development and function of the pancreas, with recent research highlighting its considerable influence on intestinal endocrine processes. Beucher A. et al. discovered an obvious reduction in the number of enterochromaffin cells, which directly impacted the secretion of 5-HT [51]. Thus, investigating the effects of FUC on the expression level of the genes encoding TRPC4 and PAX4 could further elucidate their modulation of 5-HT metabolism. Study carried out by Zhao J. et al. indicated that Liu-Jun-Zi (traditional Chinese prescription) could alleviate visceral hypersensitivity by decreasing the expression of genes encoding TRPC4 and PAX4 [52]. Our findings diverged from Zhao J. et al.’s perspective. Specifically, we observed the upregulation of TRPC4 and PAX4 in the hippocampus rather than the duodenum. Additionally, FUC exhibited efficacy in reversing these alterations, which support our finding that FUC may mitigate visceral hypersensitivity by modulating the brain–gut axis. However, our examination of most transporters and channel proteins was limited to the mRNA level, and future investigations should aim to assess their expression at the protein level to validate our findings.

Our pervious study unveiled the remarkable beneficial effects of FUC on FD mice with gastric dysmotility [25]. According to the results of 16S-rRNA sequencing, IA administration led to an obvious imbalance of gut microbial diversity and abundance. Amarl F. et al. demonstrated that specific symbiotic bacteria are crucial for the development of pain sensitivity in mice [53]. Separate study verified above opinion by employing transplantation of fecal microbiota. After transplanting fecal flora from IBS patients to germfree rats, these rats developed various IBS-like symptoms including increased sensitivity to colorectal distention [54]. Moreover, several clinical trials have confirmed significant differences in the intestinal flora structure between patients with FGIDs characterized by visceral hypersensitivity and healthy volunteers [55]. In current study, oral IA induced alterations in α-diversity and β-diversity, which is consistent with Chen M. et al.’s results [56]. However, FUC did not recover the disordered α-diversity according to the results of the Ace, Chao, and Sobs, which may be due to the short intervention period in this study. Encouragingly, the results of the PLS-DA show that FUC effectively mitigated the abnormal β-diversity induced by IA, especially oral FUCH. These results fully demonstrate the potential of FUC in regulating the gut microenvironment.

In addition, FUC exhibited mild regulatory effects on the gut microbial composition. At the phylum level, IA administration decreased the proportion of Firmicutes/Bacteroidota, especially the level of Firmicutes. A recent study by Hu L. et al. reported similar findings, demonstrating that circadian dysregulation induced visceral hypersensitivity and injured intestinal barrier and reduced the abundance of Firmicutes [57]. In the current study, FUC obviously increased the richness of Firmicutes. Li X. et al. recently uncovered this capability of FUC, showing its efficacy in diminishing inflammation levels in mice with colorectal cancer. Moreover, they elucidated that FUC can normalize the proportion of Firmicutes/Bacteroidota by augmenting Firmicutes abundance. Additionally, our investigation revealed a significant reduction in Patescibacteria at the phylum level. Studies conducted by Shi L. and Li Y. et al. reported reduced Patescibacteria in the gut microbiota of mice with either liver injury induced by perfluorooctanoic acid or obesity induced by high-fat diet [58,59]. In addition, similar results were observed in the gut flora of mice with inflammation-induced colon cancer [60]. These results suggest a potential association between decreased richness in Patescibacteria and chronic inflammation and oxidative stress. Moreover, Patescibacteria, a strict anaerobic bacterium, lack complete biosynthetic and DNA repair capabilities, relying mainly on ectosymbiosis for survival in the gut environment [61]. The reduction in Patescibacteria abundance may, thus, be linked to intestinal mucosal barrier damage. Although the effects of FUC on Patescibacteria remain uninvestigated, based on the existing literature, we suggest that FUC may elevate the abundance of Patescibacteria owing to its notable anti-inflammatory and antioxidant properties. At the genus level, FUC intervention surprisingly downregulated the abundance of *Muribaculaceae*. While the changes in *Muribaculaceae* under visceral hypersensitivity remain inconclusive, it is plausible that chronic inflammation induced by IA contributes to the decline in the abundance of *Muribaculaceae* [62]. In our previous research, FUC administration reduced the richness in *Muribaculaceae* of mice with gastric dysmotility, which was different with the findings in the current study. The association between the abundance of *Muribaculaceae*, visceral hypersensitivity, and FUC need to be explored in greater depth.

The correlation between the gut microbiota composition and biochemical indexes provided more evidence for the modulatory effects of FUC on 5-HT metabolism and microbial profile. It is noteworthy that the richness in Firmicutes was negatively associated with the secretion of COL and CORT. In research conducted by Takahashi E. et al., they found that the reduced Firmicutes was accompanying elevated serum CORT in depressive mice [63]. Similar results were concluded by Li F. et al. recently [64]. Our findings confirm their opinions and suggest the potential link between Firmicutes and depressive-related hormones. At the genus level, serum COL and CORT were related to the abundance of many bacteria, which emphasizes the changes in the microbial pattern under visceral hypersensitivity induced by stress. These conclusions support our hypothesis that FUC may relieve visceral hypersensitivity via influencing microbiota and 5-HT metabolism. However, the current study has certain limitations. Firstly, FD patients with visceral hypersensitivity often experience dysfunction of the brain–gut axis. Our animal model effectively simulates brain–gut axis dysfunction, but it does not fully capture the complexities of pain perception during visceral hypersensitivity. Secondly, although some studies have reported on the effects of FUC on microbial metabolites, these effects may vary across different pathological states, warranting further investigation. Thirdly, in the current study, we only investigated the effect of a short-term intervention of FUC on FD induced by IA, the long-term efficacy remains to be determined. In addition, while our findings suggest potential associations between 5-HT metabolism and the gut microbiota, further research is necessary to further confirm whether FUC can regulate the 5-HT–gut-microbiota axis and to elucidate the underlying mechanisms.

## 4. Materials and Methods

### 4.1. Materials and Reagents

FUC sourced from *Laminaria japonica* was procured from Yuanye Biological Co., Ltd., Shanghai, China. FUC’s molecular structure was determined via the high-performance gel permeation chromatography (HPGPC) method. The molecular weight (Mw) of fucoidan was approximately 419.6 kDa, with an Mw/Mn ratio of 1.69. The monosaccharide composition was analyzed via gas chromatography (GC), with detailed findings reported in our prior study [25]. IA was procured from Aladdin Biological Co., Ltd. (Shanghai, China).

### 4.2. Animal Experiment: Grouping Design and Intervention Strategy

#### 4.2.1. Animal Maintenance

Thirty male BALB/C mice, weighing 22 ± 2 g, were selected as the experimental subjects which were provided by the Experimental Animal Center of Huazhong Agricultural University (Wuhan, China). The animal license number is SYXK (E) 2020-0084. Experimental animals were placed in an environmental control room with specific pathogen-free (SPF) level. Mice were maintained in a temperature-controlled facility (24 ± 2 °C, relative humidity of 40 to 70%) with a 12 h light–dark cycle (8:00 to 20:00 every day). All experimental procedures and animal welfare practices strictly adhered to the National Research Council’s Guide for the Care and Use of Laboratory Animals (Eighth Edition), as well as the ethical regulations established by Huazhong Agricultural University. This study was approved by the Institutional Animal Care and Use Committee of Huazhong Agricultural University (approval ID: HZAUMO-2023-0221).

#### 4.2.2. Animal Treatment

After mice were acclimatized for 7 days, all mice were randomly assigned to one of five groups (6 mice per group) with similar body weight. The groups were as follows: (1) CK group; (2) IA-induced FD-MC groups, which received an oral intervention of 0.1 mL/kg saline; (3) FUCL and (4) FUCH intervention groups, which were orally administered 100 and 200 mg/kg FUC, respectively (saline solution) according to the protocol described in our previous experiment [25] and others’ studies [65,66]; and (5) the pinaverium bromide (MCP) treatment group positive control), which received oral administration of 5 mg/kg pinaverium bromide. The intervention dosage of FUC was equivalent to, approximately, daily doses of 567 and 1134 mg/70 kg body weight. The above oral administrations were conducted daily for 14 days. In addition, mice in the CK group were allowed access to normal drinking water. Over a 7-day period, mice in the other groups were treated with 0.1% IA in a 2% sucrose solution. From the 8th day, mice in all groups were provided with normal drinking water for 7 days (refer to Figure 1A). During the animal trial, the changes in body weight, food intake, and water intake of the animals were monitored and recorded.

### 4.3. Serum Biochemical Analysis

The animals were euthanized, and blood samples were collected by removing the eyeballs. The concentrations of 5-HT, ACHT, COL, and CORT in serum were evaluated using an ELISA Kit (Huding Biological Co., Ltd., Shanghai, China). The experimental procedures were performed in accordance with the instructions.

### 4.4. Immunohistochemistry and Immunofluorescence Staining

Immunohistochemistry and immunofluorescence staining were employed to assess the expression of 5-HT_3_R protein in stomach tissue and brain, respectively. The protocols were carried out following the previous studies [25,67]. Briefly, the detection was conducted against 5-HT_3_R (Sanying Biological Co., Ltd., Wuhan, Hubei, China). All sections were incubated with the corresponding IgG and imaged using a microscopy system. (Nikon Eclipse C1, Upright Optical Microscope, Nikon, Tokyo, Japan). An analysis of the area density and fluorescence signal was performed using Alpathwell. (Servicebio Biologial Co., Ltd., Wuhan, Hubei, China).

### 4.5. Quantitative Real-Time PCR

RNA was extracted from stomach, duodenum, hippocampus, and hypothalamus tissues using Trizol reagent (R21086, Sopo Biological Technology Co., Ltd., Guangzhou, China) [25]. The cDNA was synthesized using the Takara PrimeScript RT reagent Kit. (RR047A, Takara Bio Inc., Tokyo, Japan). Real-time PCR was performed using iTaq Universal SYBR Green Supermix Kit (Bio-Rad Laboratories, Inc., Hercules, CA, USA). The primers were synthesized by Sangon Biotech Co., Ltd., Shanghai, China. Gene expression data were analyzed using the Bio-Rad Thermal Cycler (Hercules, CA, USA). The reaction parameters were set as follows: one cycle at 95 °C for 3 min and then 39 cycles at 95 °C for 10 s and at 55 °C for 30 s. The primer details are provided in Table 1 (SERT, TPH1, TPH2, TRPC4, and PAX4).

### 4.6. Western Blot

Total protein was extracted from the duodenum using RIPA lysis buffer (Elabscience Co., Ltd., Wuhan, Hubei, China). The proteins (10–15 μL, 30 μg) were subjected to 10% SDS-PAGE for 3 h, and then the resolved proteins were transferred to nitrocellulose membranes for 2 h at 250 V. The above membranes were incubated with primary antibody overnight at 4 °C. (Sanying Co. Ltd., Wuhan, Hubei, China). After washing with TBST buffer, the membranes were incubated with 1:3000 goat anti-rabbit IgG for 3 h (Sanying Co. Ltd., Wuhan, Hubei, China). Subsequently, ECL luminescent solution was added to the membranes (SuperKine™, Abbkine Scientific Co., Ltd., Wuhan, Hubei, China). The images of each protein were acquired using Odyssey FC system (LI-COR, Inc., Lincoln, NE, USA). The density of protein quantified using Image J (1.53 t).

### 4.7. Gut Microbiota Analysis by 16S rRNA Gene Sequencing

All experiments were conducted by Majorbio Co., Ltd., Shanghai, China. Briefly, The fecal microbial community was determined using Illumina Next-Generation 16S rRNA gene amplicon sequencing, which have been described in previous studies [25,68]. Sequences with ≥97% similarity were clustered into OTUs using Uparse software (Version 7.0.1001). Microbial diversity was analyzed using the R programming language (Version. 3.3.1). The richness and diversity of the microbial communities were analyzed by the α-diversity. The similarities and differences of each sample were analyzed by the β-diversity.

### 4.8. Statistical Analysis

All data in this study are expressed as the mean ± standard deviation (SD). One-way analysis of variance (ANOVA) was used for the analysis of the statistical significance and followed by post hoc multiple comparisons with the Tukey test and Duncan test. (SPSS R26.0.0.0, Chicago, IL, USA). A *p*-value < 0.05 or <0.01 was considered statistically significant.

## 5. Conclusions

As a prevalent FGID, FD still lacks multi-target medicines with long-term efficacy. Herbal/dietary extracts have potential as ingredients in complementary and alternative medicine. According to previous evidence, we speculated FUC may ameliorate EPS-FD characterized by visceral hypersensitivity. After a detailed assessment of indicators at different levels, it can be concluded that oral FUC can mitigate visceral hypersensitivity through modulating depression-related hormones, 5-HT metabolism, expression of 5-HT_3_R, and the gut microbiota profile. The above results indicate that FUC has remarkable potential in regulating visceral sensory disorder by influencing 5-HT metabolism and gut microbiota. However, more experiments are necessary to explore and confirm the potential mechanisms of the above effects. Our findings provide a theoretical basis for the prospective utilization of FUC as a complementary and alternative therapy for FD.

## Figures and Tables

**Figure 1 ijms-26-03273-f001:**
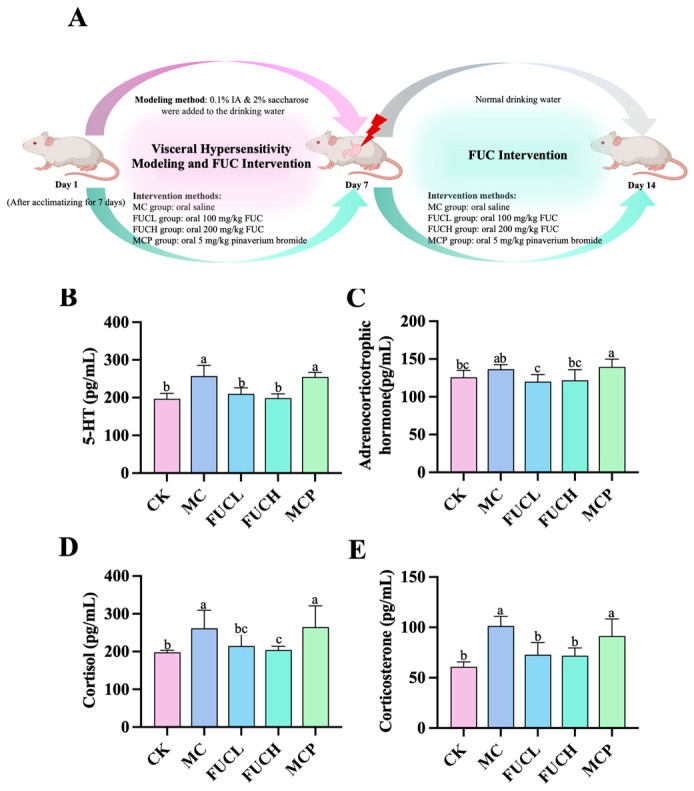
Levels of serum signaling neurotransmitter and hormones in mice for each group after the intervention: the intervention protocol for the current study (**A**); effect of FUC on the levels of 5-HT (**B**), ACHT (**C**), COL (**D**), and CORT (**E**) (n = 5–6). Columns labeled with different abbreviations represent statistical significance (*p* < 0.05). IA administration significantly increased the levels of 5-HT, COL, and CORT. The intervention of FUC reduced the secretion of 5-HT, ACHT, COL, and CORT. ACHT, adrenocorticotrophic hormone; COL, cortisol; CORT, corticosterone; FUC, fucoidan; FUCL, low-dose fucoidan; FUCH, high-dose fucoidan; IA, iodoacetamide; 5-HT, 5-hydroxytryptamine.

**Figure 2 ijms-26-03273-f002:**
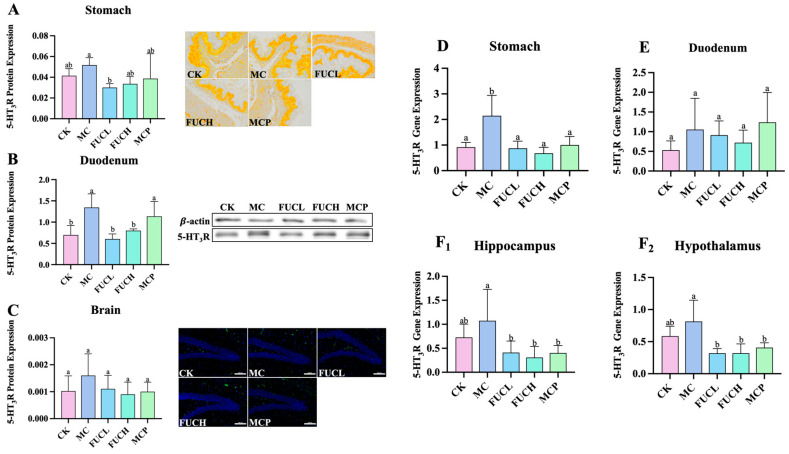
Expression of 5-HT_3_R at the protein and mRNA levels. Effect of FUC on 5HT_3_R expression in stomach tissues ((**A**), 200×), duodenum tissues (**B**), and brain ((**C**), 200×) at the protein level (n = 4–6). At the mRNA level, the expression of the gene encoding 5-HT_3_R was determined in the stomach (**D**), duodenum (**E**), hippocampus (**F_1_**), and hypothalamus (**F_2_**). Columns labeled with different abbreviations represent statistical significance (*p* < 0.05). Oral FUC reduced the 5-HT_3_R receptor in stomach and duodenum tissues at the protein level. Meanwhile, the expression of the gene encoding 5-HT_3_R decreased in the stomach, hippocampus, and hypothalamus after FUC administration. FUC, fucoidan; 5-HT_3_R, 5-HT_3_ receptor.

**Figure 3 ijms-26-03273-f003:**
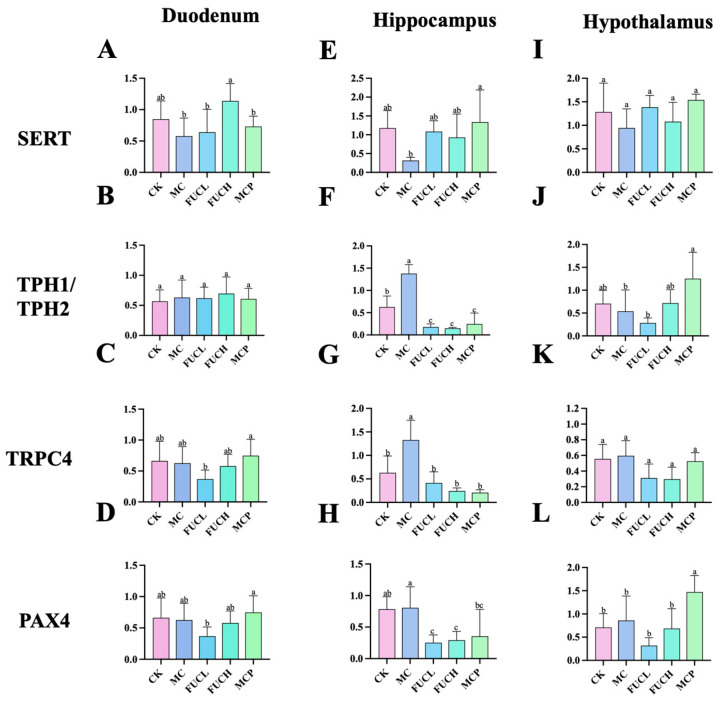
Expression of key genes in mice for each group in the duodenum, hippocampus, and hypothalamus. The effect of FUC on the expression of the genes encoding of SERT (**A**,**E**,**I**), TPH1/TPH2 (**B**,**F**,**J**), TRPC4 (**C**,**G**,**K**), and PAX4 (**D**,**H**,**L**) (n = 5–6). Columns labeled with different abbreviations represent statistical significance (*p* < 0.05). Major changes were concentrated in the hippocampus. Oral FUC reversed the disordered expression of these kinds of gene. FUC, fucoidan; PAX4, paired box 4; SERT, serotonin transporter; TPH1/TPH2, tryptophan hydroxylase 1/2.

**Figure 4 ijms-26-03273-f004:**
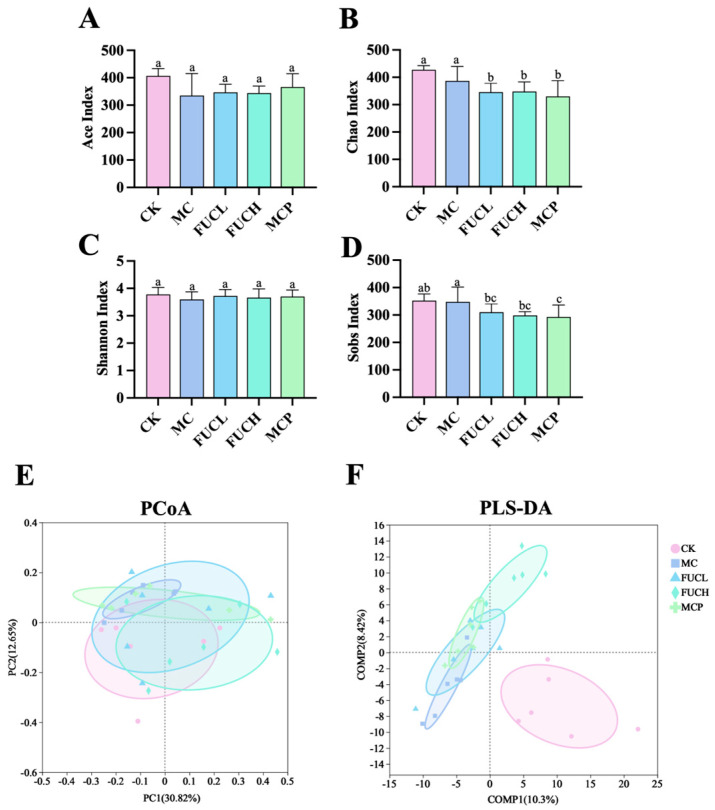
The α-diversity and β-diversity of the gut microbial community after intervention in each group. The Ace (**A**), Chao (**B**), Shannon (**C**), and Sobs (**D**) indexes, the PCoA and PLS-DA plots of the gut microbiota at the OTU (**E**,**F**) level (n = 5–6). Columns labeled with different abbreviations represent statistical significance (*p* < 0.05). IA administration led to an imbalance in the gut microbial diversity, while oral FUC partly reversed the changes. FUC, fucoidan; IA, iodoacetamide.

**Figure 5 ijms-26-03273-f005:**
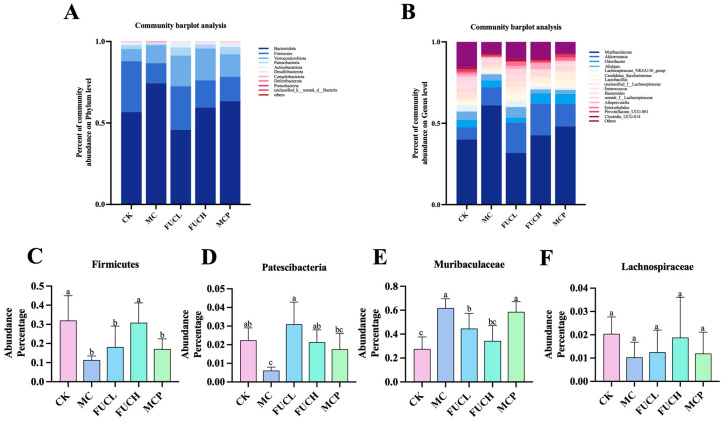
The profile of gut microbiota after intervention in each group. Comparison of the compositions of the different gastrointestinal tract microbiomes at the phylum level (**A**) and the genus level (**B**). Representative difference in bacteria at the phylum (**C**,**D**) and genus levels (**E**,**F**) (n = 5–6). Columns labeled with different abbreviations represent statistical significance (*p* < 0.05). Compared to the CK group, the mice in the MC group had an obviously different gut microbial profile. The intervention of FUC mitigated the gut microbial abnormality induced by IA. CK, control check; FUC, fucoidan; MC, model control.

**Figure 6 ijms-26-03273-f006:**
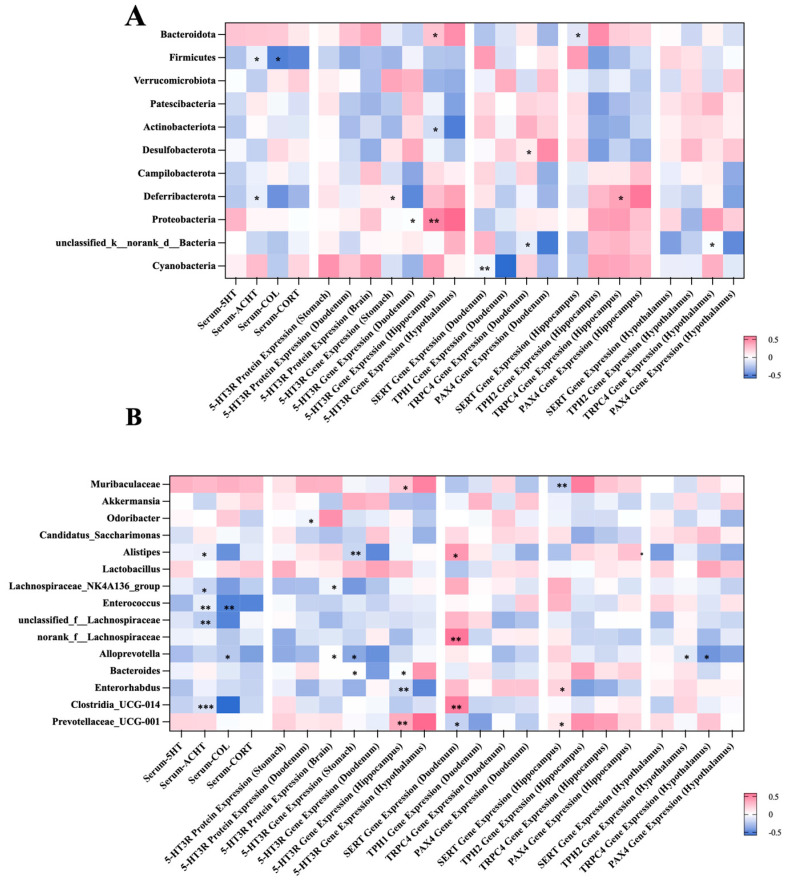
Spearman’s correlation analysis between the bacteria genera and gastrointestinal-dysmotility indicators at the phylum level (**A**) and the genus level (**B**). Significant correlations are marked as the following: * *p* < 0.05, ** *p*
< 0.01, and *** *p* < 0.001. At the phylum level, the abundance of Firmicutes was negatively associated with the levels of serum COL and CORT. The level of serum COL was negatively related to the richness of *Alistipe*, *Lachnospiraceae_NK4A136_group*. COL, cortisol; CORT, corticosterone.

**Table 1 ijms-26-03273-t001:** Summary of the primer sequences.

Gene	Primer Sequence
*5-HT3R*	(5′-3′)-forward GCTATCCTCCATCCGCCACTTC
	(5′-3′)-reverse CGAGCACAGCCAGCAGGTAG
*5-HT4R*	(5′-3′)-forward GCTATCCTCCATCCGCCACTTC
	(5′-3′)-reverse CGAGCACAGCCAGCAGGTAG
*SERT*	(5′-3′)-forward ATGGCTGAGATGAGGAACGAAGAC
	(5′-3′)-reverse AAGAATGTGGATGCTGGCATGTTAG
*TPH1*	(5′-3′)-forward GGCTGAGGATGCGTCAACATTAAC
	(5′-3′)-reverse CCTCTTAGTCGCTGTCTGCTGTC
*TPH2*	(5′-3′)-forward CCTGATGTGGCAATGACCTAAGTG
	(5′-3′)-reverse CAGAGACAGAGACAGAGACAGAGAG
*TRPC4*	(5′-3′)-forward AGAGCGAAGGTAATGGCAAGGAC
	(5′-3′)-reverse GCACCACCAGGGCGAAC
*PAX4*	(5′-3′)-forward CTCCTGAGTGAAGGCTCTGTGAAG
	(5′-3′)-reverse TGCTGGTGGTCTGGTCTTGTAAC
β *-actin*	(5′-3′)-forward GATGGTGGGAATGGGTCAGAAGG
	(5′-3′)-reverse TTGTAGAAGGTGTGGTGCCAGATC

## Data Availability

All relevant data are provided in the paper.

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
