# Peer review of "Protective Effects of Fucoidan on Iodoacetamide-Induced Functional Dyspepsia via Modulation of 5-HT Metabolism and Microbiota"

_ijms, 2025, doi:10.3390/ijms26073273_

Round 1

Reviewer 1 Report

Comments and Suggestions for Authors

   In this paper, the authors investigated the regulatory effects of fucoidan on visceral hypersensitivity, focusing on its impact on intestinal microbiota and 5-HT metabolism. This paper presents certain innovation and is supported by a substantial amount of experimental data. It provides valuable evidence for the ability of fucoidan in improving gastrointestinal health.

   I have the following concerns and suggestions:

  1. Authors does not clearly explain the rationale for selecting doses of 100 mg/kg and 200 mg/kg of fucoidan (e.g., whether these doses were based on preliminary experiments or effective dose ranges reported in the literature). It is recommended to provide background information for the dose selection. Besides, I suggest that authors should added the experimental process to the main text Fig.
  2. Can IA-induced FD model fully replicate the complex symptoms of clinical FD, such as brain-gut axis dysfunction? It is recommended to compare this model with clinical FD patients in the discussion.
  3. The study only employed a 14-day intervention period. The authors did not investigate the effects of long-term intervention of IA and fucodian. It is necessary to address the limitations of short-term intervention.
  4. Some experimental results appear to be not statistically significant across all groups, particularly the gut microbiota data.
  5. The language needs to be polished carefully.

The overall quality of the manuscript is acceptable, I recommend a major revision.

Comments on the Quality of English Language
  1. The language needs to be polished carefully.

Author Response

Responds to the reviewer #1’s comments:

Response to comment 1: Thank you very much for your comments. The dosage employed in current study is an accepted in vivo intervention dose, as previously established in our experiment and other studies. We added relevant content in line 423-424 and 426-427. Besides, we added a figure to describe the experimental process (Figure 1).

Response to comment 2: Thank you for your question. The advantages of IA-induced FD model have been explained in line 241-243. The animal model of visceral hypersensitivity induced by IA has been widely employed in previous studies1. The non-organic gastrointestinal inflammation and depression-like behavior are consistent with certain clinical symptoms of FD.

Response to comment 3: Thank you very much for your patient review. We added relevant content was added in line 390-391.

Response to comment 4: We really appreciate your kind comment. In response to your question, I would like to make the following explanation. The gastrointestinal disorder discussed in this paper, namely FD, is classified as a non-organic disease, signifying only dysfunction and the absence of organic changes in patients with FD. Compared to organic digestive disease, such as peptic ulcer or inflammatory bowel disease, there are no definite hematological and histological indicators for the diagnosis of FD. Therefore, compared to CK (control check) group, there was no statistical difference in some indicators (key genes in stomach and duodenum, etc.) in the MC (iodoacetamide-induced visceral hypersensitivity-model control) group. Meanwhile, as a drug commonly used in the clinical treatment of FD, pinaverium bromide is effective but did not show statistically significant changes in the related indexes of partial visceral hypersensitivity. Fucoidan had a certain effect on FD and had a regulating effect on visceral hypersensitivity. Although the above regulatory effects did not show statistical differences, these changes are generally considered to be effective given the non-organic nature of most functional diseases. We added some description in line 42-44.

Reviewer 2 Report

Comments and Suggestions for Authors

The authors of this study investigated the regulatory effects and underlying mechanisms of fucoidan on functional dyspepsia with visceral hypersensitivity. The article presents several innovative aspects, with a well-structured experimental design and adequate expeiment, providing a basis for revealing the biological activity of fucoidan. I think the article is suitable for publication in International Journal of Molecular Sciences. However, some problems need to be solved before publication, as follows:

  1. In the Introduction section, the authors should restructure and refine the content to enhance its fluency.
  2. Based on your previous research, which investigated the regulatory effects of Laminaria on visceral hypersensitivity, what prompted you to explore the effects of fucoidan in the current study?
  3. Does the intervention dose of fucoidan used in this study have a heoretical basis? Have similar doses been employed in previous studies?
  4. In Result 3.2, the statistical difference between the CK and MC groups appears to be non-significant (Figure 2C, 2E, 2F1, 2F2). Could you please explain the reasons for this?
  5. The Result 3.4 and Discussion should provide a more detailed explanation of the effects of fucoidan on the intestinal microbiota.
  6. The overall format of the article should be carefully reviewed, including the formatting of the references (such as ref 11, 13, 17, etc.).
Comments on the Quality of English Language

English expression and grammar need further careful examination.

Author Response

Responds to the reviewer #2’s comments:

Response to comment 1: Thank you very much for your advice. The structure of introduction has been refined in line 42-44, 53-60, 72-76.

Response to comment 2: Thank you for your question. Fucoidan and laminarin are two kinds of polysaccharides extracted from brown algae. We suggest that polysaccharides from marine plants have broad and powerful biological activities. Although the structures of fucoidan and laminarin are not similar, polysaccharides with different structures derived from brown algae may exhibit similar biological functions. Comparing their ameliorative effects on the same dysfunction is valuable for a comprehensive understanding of the biological functions of brown algal polysaccharides. Furthermore, their synergistic effects will be deeply explored in our future studies.

Response to comment 3: Thank you very much for your comments. The dosage employed in current study is an accepted in vivo intervention dose, as previously established in our experiment and other studies. We added relevant content in line 423-424 and 426-427.

Response to comment 4: We really appreciate your kind comment. In response to your question, I would like to make the following explanation. The gastrointestinal disorder discussed in this paper, namely FD, is classified as a non-organic disease, signifying the absence of organic changes in patients with FD. Compared to other digestive disease, such as peptic ulcer or inflammatory bowel disease, there are no definite hematological and histological indicators for the diagnosis of FD. Therefore, compared to CK (control check) group, there was no statistical difference in some indicators in the MC (iodoacetamide-induced visceral hypersensitivity-model control) group. Meanwhile, as a drug commonly used in the clinical treatment of FD, pinaverium bromide is effective but did not show statistically significant changes in the related indexes of partial visceral hypersensitivity. Fucoidan had a certain effect on FD and had a regulating effect on visceral hypersensitivity. Although the above regulatory effects did not show statistical differences, these changes are generally considered to be effective given the non-organic nature of most functional diseases. We added some description in line 42-44.

Response to comment 5: Thank you for your advice. We added and modified some content in line 186, 192, 364-365, and 370-373.

Response to comment 6: We appreciate your patient review. We carefully checked the format in this article.

Reviewer 3 Report

Comments and Suggestions for Authors
  1. Please provide relevant assessment data on gastrointestinal function such as gastric emptying time, smooth muscle contraction ability and digestive juice secretion ability, etc..
  2. From the results, Iodoacetamide elevated serum 5-HT, cortisol and corticosterone levels. How are these related to functional dyspepsia? Authors must provide the 5-HT levels in different regions.
  3. Why different methods are used to analyze 5-HT3R protein expression in different regions.
  4. Iodoacetamide only elevated the 5-HT3R protein expression in duodenum, however it only elevated the 5-HT3R mRNA expression in the stomach. Why?
  5. Iodoacetamide elevated the 5-HT3R mRNA expression in the stomach. Why not analyze the key genes of stomach tissue?
  6. Iodoacetamide did not alter the expression of selected key genes, however, it elevated the 5-HT3R protein expression in duodenum. Why?
  7. What is the causal relationship between changes in 5-HT levels and changes in gut microbial community? What is the exact mechanism by which Iodoacetamide causes functional dyspepsia?
  8. Please provide the timeline figure for experimental design.
  9. Please provide a diagram of the possible mechanism of Iodoacetamide and the point of action of FUC.

Author Response

Responds to the reviewer #3’s comments:

Response to comment 1: Thank you so much for your question. The regulatory effects of fucoidan on gastric motility (including gastric emptying time, smooth muscle contraction ability, etc.) have been investigated in our previous study2. In current study, we mainly explored the regulatory ability of fucoidan on FD with visceral hypersensitivity. Visceral hypersensitivity and gastric motility are the two main causes of FD, which cannot be fully investigated using a single animal model or study.

Response to comment 2: We appreciate your patient review. 5-HT, also called serotonin, is a vital gastrointestinal neurotransmitter which responsible for transmitting information and sensation between nerve cells. The abnormal metabolism of 5-HT play an important role in FD with visceral hypersensitivity. Besides, ACHT, COL, and CORT are secreted by pituitary and adrenal glands, the responsiveness of hypothalamic-pituitary-adrenal axis usually increases during states of stress. The associations of 5-HT, cortisol, and corticosterone and visceral hypersensitivity have been investigated and discussed by many studies3,4. Relevant content has been discussed in line 238-249. In our study, we focused on serum 5-HT levels because it serves as a systemic biomarker that is closely associated with the pathophysiology of functional dyspepsia. Previous studies have shown that serum 5-HT levels can reflect the overall gut-brain axis dysfunction5,6, which is a key mechanism in functional dyspepsia.

Response to comment 3: Thank you so much for your careful review. We employed immunofluorescence, immunohistochemistry, and Western blotting to comprehensively assess the expression of the same receptor (5-HT3R) in the brain, stomach, and duodenum. Each method offers unique advantages: Immunofluorescence enables cellular localization, making it ideal for observing the expression of receptors across different brain regions. Additionally, due to the small size of the mouse brain, it is challenging to extract sufficient protein for Western blot analysis. Besides, we felt sorry for our lack of rigor. The detection of 5-HT3R in stomach was conducted earlier, which is why immunohistochemistry was used instead of Western blot analysis. We will ensure more careful consideration of this issue in future experiments.

Response to comment 4: Thank you for this question. The differential effects of iodoacetamide on 5-HT3R protein expression in the duodenum and mRNA expression in the stomach may be due to tissue-specific regulatory mechanisms. In the duodenum, iodoacetamide could enhance protein expression by affecting translation or post-translational modifications. In contrast, in the stomach, the observed elevation in mRNA expression may result from transcriptional regulation or an increased mRNA stability without corresponding changes in protein synthesis. These tissue-specific differences could reflect distinct cellular responses to iodoacetamide or the presence of different regulatory factors in each tissue. The specific reasons are worthy of further studies.

Response to comment 5: Our study focused on analyzing the expression of key genes (SERT, TPH1, etc.) in the duodenum because this tissue is directly involved in the pathophysiology of FD, particularly in relation to gut-brain signaling7,8. While the stomach is important in FD, our model specifically targeted changes in the duodenum, as it plays a crucial role in the gut's interaction with the central nervous system. We did not analyze key genes expression in the stomach in current experiment, as the primary aim was to understand the duodenal changes in relation to 5-HT3R expression, which is a key focus of our study. Future studies may consider extending this analysis to the stomach tissue for a more comprehensive understanding.

Response to comment 6: Thank you so much for your professional question. The increase in 5-HT3R protein expression does not necessarily indicate a disruption in 5-HT metabolism, as receptor expression can be regulated independently of the genes involved in serotonin synthesis and reuptake, such as SERT, TPH1, TRPC4, and PAX4. While these key genes are closely related to 5-HT metabolism, FD is a non-organic disorder, where changes in gene expression may not always be significant. The elevation in 5-HT3R protein expression could reflect a compensatory or receptor-specific alteration without a corresponding change in the expression of 5-HT-related genes.

Response to comment 7: The mild modeling approach we employed, involving the add of 0.1% IA to the drinking water for a short intervention period, likely did not contributed to the significant changes in the -diversity of the intestinal microbiota. According to figure 4F and figure 5, it can be concluded that IA induced obvious imbalance of microbial composition, which may be one of the reasons by which IA causes FD. The correlations between 5-HT and microbiota have been analyzed in figure 6. Our results indicated the relationship between microbial composition and 5-HT at the phylum and genus levels. We will further explore the relationship between specific microorganism and 5-HT metabolism through in vivo and in vitro experiments.

Response to comment 8: Thank you for your suggestions. A timeline figure was added as figure 1A.

Response to comment 9: Thanks for your suggestion. The diagram of possible mechanisms was shown in the graphical abstract.

Reference

  1. Bai Y, Liu F, Wan Y, et al. Network pharmacology combined with experimental validation reveals the mechanism of action of erpixing granules on functional dyspepsia. J Ethnopharmacol. Nov 15 2024;334:118553. doi:10.1016/j.jep.2024.118553
  2. Liu T, Zhang M, Asif IM, Wu Y, Li B, Wang L. The regulatory effects of fucoidan and laminarin on functional dyspepsia mice induced by loperamide. Food Funct. Jul 17 2023;14(14):6513-6525. doi:10.1039/d3fo00936j
  3. Greenwood‐Van Meerveld B, Moloney R, Johnson A, Vicario M. Mechanisms of stress‐induced visceral pain: implications in irritable Bowel syndrome. Journal of neuroendocrinology. 2016;28(8)
  4. Grundy D. 5-HT system in the gut: roles in the regulation of visceral sensitivity and motor functions. European Review for Medical & Pharmacological Sciences. 2008;12
  5. Chen SH, Wu HS, Jiang XF, et al. Bioinformatics and LC-QTOF-MS based discovery of pharmacodynamic and Q-markers of Pitongshu against functional dyspepsia. J Ethnopharmacol. Jul 15 2024;329:118096. doi:10.1016/j.jep.2024.118096
  6. Guo J, Chen L, Wang YH, et al. Electroacupuncture Attenuates Post-Inflammatory IBS-Associated Visceral and Somatic Hypersensitivity and Correlates With the Regulatory Mechanism of Epac1-Piezo2 Axis. Front Endocrinol (Lausanne). 2022;13:918652. doi:10.3389/fendo.2022.918652
  7. Jung HK, Talley NJ. Role of the Duodenum in the Pathogenesis of Functional Dyspepsia: A Paradigm Shift. J Neurogastroenterol Motil. Jul 30 2018;24(3):345-354. doi:10.5056/jnm18060
  8. Wauters L, Ceulemans M, Vanuytsel T. Duodenum at a crossroads: Key integrator of overlapping and psychological symptoms in functional dyspepsia? Neurogastroenterol Motil. Oct 2021;33(10):e14262. doi:10.1111/nmo.14262

Round 2

Reviewer 1 Report

Comments and Suggestions for Authors

The authors have adequately addressed the concerns raised and the quality of the manuscript has been improved after the revisions.I do not have any additional comments or queries.